# Cross-Talk between Inflammatory Mediators and the Epithelial Mesenchymal Transition Process in the Development of Thyroid Carcinoma

**DOI:** 10.3390/ijms20102466

**Published:** 2019-05-18

**Authors:** Giovanna Revilla, Rosa Corcoy, Antonio Moral, Joan Carles Escolà-Gil, Eugenia Mato

**Affiliations:** 1Institut d’Investigacions Biomèdiques (IIB) Sant Pau, Hospital de la Santa Creu i Sant Pau (HSCSP), 08041 Barcelona, Spain; giovannarevilla@hotmail.com; 2Departament de Bioquímica, Biologia Molecular i Biomedicina, Universitat Autònoma de Barcelona (UAB), 08193 Barcelona, Spain; 3Department of Endocrinology, Hospital de la Santa Creu i Sant Pau (HSCSP), 08025 Barcelona, Spain; rcorcoy@santpau.cat; 4Centro de Investigación Biomédica en Red en Bioingeniería, Biomateriales y Nanomedicina (CIBER-BBN), 28029 Madrid, Spain; 5Departament de Medicina, Universitat Autònoma de Barcelona (UAB), 08193 Barcelona, Spain; 6Department of General Surgery—Hospital de la Santa Creu i Sant Pau (HSCSP), 08025 Barcelona, Spain; amoral@santpau.cat; 7Departament de Cirugia, Universitat Autònoma de Barcelona (UAB), 08193 Barcelona, Spain; 8Centro de Investigación Biomédica en Red en Diabetes y Enfermedades Metabólicas Asociadas (CIBERDEM), 28029 Madrid, Spain

**Keywords:** thyroid carcinoma, signaling pathways, MET/EMT, leptin, adiponectin, low-density lipoproteins, high-density lipoproteins

## Abstract

There is strong association between inflammatory processes and their main metabolic mediators, such as leptin, adiponectin secretion, and low/high-density lipoproteins, with the cancer risk and aggressive behavior of solid tumors. In this scenario, cancer cells (CCs) and cancer stem cells (CSCs) have important roles. These cellular populations, which come from differentiated cells and progenitor stem cells, have increased metabolic requirements when it comes to maintaining or expanding the tumors, and they serve as links to some inflammatory mediators. Although the molecular mechanisms that are involved in these associations remain unclear, the two following cellular pathways have been suggested: 1) the mesenchymal-epithelial transition (MET) process, which permits the differentiation of adult stem cells throughout the acquisition of cell polarity and the adhesion to epithelia, as well to new cellular lineages (CSCs); and, 2) a reverse process, termed the epithelial-mesenchymal transition (EMT), where, in pathophysiological conditions (tissue injury, inflammatory process, and oxidative stress), the differentiated cells can acquire a multipotent stem cell-like phenotype. The molecular mechanisms that regulate both EMT and MET are complex and poorly understood. Especially, in the thyroid gland, little is known regarding MET/EMT and the role of CCs or CSCs, providing an exciting, new area of knowledge to be investigated. This article reviews the progress to date in research on the role of inflammatory mediators and metabolic reprogramming during the carcinogenesis process of the thyroid gland and the EMT pathways.

## 1. Introduction

Epithelial thyroid cancer (ETC) is the most common malignancy of the endocrine system, which affects the follicular cells of the gland. These tumors are twice as common in women as in men, but male tumors are associated with a poor prognosis. Additional important risk factors include a history of childhood head and neck irradiation, large nodule size (>4 cm), evidence for local tumor fixation or invasion into lymph nodes, and the presence of metastases. Moreover, this neoplasia increases with the age, plateauing after about age 50 years; it represents an important prognostic factor for thyroid cancer at a young age (<20), or in older persons (>45), and it is associated with a worse prognosis [1].

There are two main types of ETC based on the cells’ differentiated characteristics:

Well-Differentiated thyroid cancer (WDTC) arises from thyroid follicular epithelial cells and it accounts for the overwhelming majority of thyroid cancers. Of the differentiated cancers, papillary thyroid cancer (PTC) comprises about 85% of cases when compared with about 10%, which have follicular histology, referred to as follicular thyroid cancer (FTC), and 3–5% of the type FTC correspond to the Hürthle cell or oxyphil tumors characteristic.

Although PTC and FTC have a good prognosis, in the case of PTC, some histological subtypes, such as tall cell variant, columnar cell variant, and diffuse sclerosing variant, show aggressive behavior with a worse prognosis. In FTC tumors, 11% of patients have an aggressive tumor at diagnosis [1]. All of these tumors are classified following the Staging System for Differentiated Thyroid Cancer, from the American Joint Commission on Cancer (AJCC) [2]. In the case of ETT, the age at time of diagnosis is important to define the risk of recurrence and tumoral aggressiveness, with 55 years old being the cut-off. In younger patients (≤55 age), the stages I and II are classified with a low risk, while in patients (≥55 age) the stages I and II has a low risk, whereas stage III and IV has a high risk with local a distant metastasis [3].

PTC tumors show multifocality and they are able to invade tissues near the gland; the major tumoral dissemination is through the lymphatic system, but also hematogenously (bone and lung). Most of these tumors (80%) are classified as stages I or II, with a good prognosis and only 1% of them lose the differentiated characteristic, with a fatal prognosis (stage IV).

In the case of FTC, epidemiological studies show that iodine-deficiency is a key factor in the development of these types of tumors, and most of them appear as a nodule in the gland with a slow growth before the metastatic process. The histological diagnosis sometimes is difficult, because it presents a normal follicular pattern to anaplastic form without follicles or colloid [4].

The tumors with a nodule are less aversive and present low risk for infiltrating the tissues closer or local adenopathy. Nevertheless, tumors with high risk show capsular invasion, high growing in the local structures (strap muscles or trachea), making their resectability difficult. The prognosis varies in function of the degree of invasion [5].

FTC tends to invade locally and metastasize distantly, rather than to local nodes; they are especially prone to metastasizing to the bone or lung. In a Massachusetts General Hospital series [6], one-half of the cases already exhibited metastasis when the diagnosis was originally established. The tumor and metastases often retain the ability to accumulate and hold iodide, and therefore, they are sometimes susceptible to treatment with Radioactive Iodine (RAI). In some cases, FTC presents a normal histological pattern, so it is important to analyze the capsular and/or vascular invasion, and the metastasis is the only reliable criterion of malignancy in these types of tumors [7].

Poorly differentiated carcinoma (PDTC) is an aggressive tumor that shares the characteristics of WDTC and anaplastic thyroid carcinoma (ATC). Their incidence is rare (0.23% to 2.6% of patients with ETC) and, although the hypothesis about their origin is that comes from dedifferentiated of the WDTC, some of them appear *de novo* in the patients. The diagnosis of is difficult, because they show a different histological pattern (solid, trabecular, or sclerotic) with differences in their growing. However, all of them have poor prognosis and lymph node metastasis and distant metastasis is frequent [8].

Undifferentiated thyroid carcinoma (UTC) corresponds to anaplastic thyroid carcinoma (ATC), which is diagnosed in persons over 50 years of age; the origin of these tumors is controversial; more than half of the diagnosed cases originate from well-differentiated tumors and represent 1–2% of all thyroid malignancies [9]. These tumoral cells show and aggressive behaviour with a rapid grow, and in most of cases at the moment of diagnosis, the patients show local (lymph nodes) and distant metastasis (pulmonary). Moreover, it is of note that only 10% of these tumors are resectable at the time of diagnosis and ^131^I therapy does not work, because they lack the typical differentiation feature of the thyroid gland (iodine uptake, thyroglobulin secretion, response to TSH stimulation) and are the most aggressive. For that reason, they have a poor prognosis with a low survival within 6–12 months [10].

## 2. Origin of Epithelial Thyroid Carcinoma (ETC)

### 2.1. Signaling Pathways of ETC Oncogenesis

Different common genetic alterations (gene mutation, gene amplifications, copy-number gains, gene translocations and aberrant gene methylation) have been identified in WDTC (PTC and FTC) and, in PDTC, UTC/ATC, being the most frequent: point mutations affecting oncogenes (BRAF, RAS, TP53, and CTNNB1), rearrangements affecting RET-PTC and NTRK genes. Such mutations or genetic disorders directly or indirectly affect different cell signaling pathways, including MAPKinasas or the PI3K-AKT-mTOR, which are essential for many cell biological processes [11,12]. These important findings in the knowledgement of ETC etiopathogenesis will hopefully allow physicians to properly select patients that require aggressive treatment and minimize the risk for those patients with indolent tumors, who may not even require surgery. This would maximize the cure rate and minimize complications [13].

The main signaling pathways that are involved in thyroid carcinogenesis are described below and are summarized in Figure 1.

The MAPK pathways regulate multiple cell progresses, including proliferation, differentiation, and survival. In PTC, the activation of the MAPK pathway by mutated BRAF-V600E, RAS, or RET/PTC and RTK rearrangements has been identified. In contrast, mutations that involve the PI3K/Akt pathway, such as the PI3K, Akt1, and phosphatase and tensin homolog (PTEN), are more frequently found in FTC and in de-differentiated types of thyroid cancer (Figure 1) (Table 1) [11].

The importance of the MAPK pathway in PTC is now well known. The MAPK pathway is affected by activating mutations, such as PRAF and RAS, RET-PTC, and in several cases, by ALK mutation [14]. Tumorigenic activity that is caused by MAPK leads to a wide range of secondary molecular alterations that synergize and amplify oncogenic activity, such as hypermethylation and hypomethylation, as well as the upregulation of different oncogenic proteins, including chemokines, vascular endothelial growth factor A (VEGFA), the mesenchymal–epithelial transition (MET), nuclear factor kB (NF-κB), matrix metalloproteinases (MMPs), prohibitin, vimentin, hypoxia-inducible factor α (HIFIα) prokineticin (also known as EG-VEGF or PROK1), urokinase plasminogen activator (uPA) and its receptor (uPAR), TGF-β1, and thrombospondin (TSP1). All of these proteins play an important role in the human tumorigenesis through different mechanisms that manage cell carcinogenic proliferation, growth, migration and survival, tumor angiogenesis, invasion, and metastases (Figure 1 and Table 1) [15].

Many of these proteins are the critical constituents of the unique microenvironment of the extracellular matrix; this plays an important role in the tumor pathogenesis of the thyroid, which is determined by fundamental oncoproteins, like BRAF-V600E. This microenvironment not only acts as a cell support, but it also has an important influence on the behavior of tumor cells, including their viability, proliferation, adhesion, and mortality [26]. Both tumor and stromal cells (fibroblasts, macrophages) produce proteins that form autocrine and paracrine signaling chains. For example, when activated, the MAPK pathway leads to the liberation of TSP1 in the matrix, where it interacts and modifies proteins, namely, cellular membrane receptors, integrins and non-integrins, matrix proteins, cytokines, VEGFA, and MMPs. These, in turn, activate the signaling system in thyroid cancer cells (CCs) and promote tumor progression and metastasis. Released into the extracellular matrix, the TGF-β1 cytokine is capable of activating the BRAF-V600E/MAPK pathway (Figure 1). The cytokines can create an inflamed microenvironment with the production of reactive oxygen species and oxidative stress, which stimulate the MAPK pathway, thereby elevating tumorigenic activity [27]. It seems that stromal fibroblasts mainly intervene through the expression patterns of two isoforms of fibroblast growth factor receptor type 2 (FGFR2)—FGFR2-IIIb and FGFR2-IIIc—in tumor cells, and fibroblasts express the same type of FGFR2 isoform (Figure 1) [16].

The PI3/Akt signaling pathway plays a significant role in thyroid tumor genesis. It was discovered to be associated with follicular thyroid adenoma (FTA) and FTC, while investigating Cowden syndrome, which is caused by germinal PTEN mutation. The Akt1 and Akt2 isoforms are clearly expressed and activated in thyroid cancer [28]. Studies with human thyroid tumors have suggested that the invasion and metastasis of FTC, as activated by this pathway, mainly affect activation and nuclear localization of Akt1, which coincides with the presence of Akt1 mutations in the metastatic thyroid cancers. This pathway predominates in FCT, while the MAPK pathway predominates in PTC. It is plausible that PI3K/Akt pathway hyperactivation leads to FTA to PTC conversion (Figure 1 and Table 1) [17].

The NF-κB signaling pathway has an important role in the regulation of inflammatory response related to the tumor generation; it also plays a significant part in thyroid cancer. This pathway controls the signaling processes of cellular proliferation and anti-apoptosis in tumoral cells; it also determines the upregulated expression of different oncogenic proteins, which the MAPK pathway also upregulates. In thyroid cancer, with different histological patterns that include PTCs, FTCs, and ATCs, the RET-PTC, RAS, and BRAF-V600E mutations promote the activation of the NF-κB pathway in differentiated tumors, thereby inducing dedifferentiation processes and also increasing the invasiveness (Figure 1 and Table 1) [18,19,20].

The RASSF1/MST1/FOXO3 signaling pathway is extremely important in tumor suppression and apoptosis promotion. BRAF-V600E, but not the native BRAF, directly interacts with the terminal carboxyl group of MST1, which inhibits its protein kinase activation; these results decreased the transactivation of FOXO3, a transcriptional factor that activates pro-apoptotic gene transcription, through its translocation to the nucleus. Hypermethylation of the RASSF1A promoter is associated with its silencing in thyroid cancer (Figure 1) [21]. In summary, BRAF-V600E is independently linked with three principle pathways, namely MEK/MAPK, RASSF1/MST1/FOXO3, and NF-κB, thereby determining a unique and powerful oncogenic mechanism in the process of thyroid tumorigenesis (Table 1).

The WNT/β-catenin signaling pathway plays a well-established role in cell growth and proliferation, as well as in the differentiation of cell progenitors, and its constitutive activation is frequently seen in the oncogenic processes. β-catenin facilitates the transcription of diverse tumor-promoting genes through its upregulation by the WNT signaling pathway [29]. In the case of thyroid cancer, the activation of this pathway is determined by the activating mutation of CTNNB1 (which codifies β-catenin), especially in terms of ATC and PDTC (Figure 1) (Table 1) [22].

In terms of the HIF1α signaling pathway, it is well-known that hypoxia is a strong stimulus for tumor metabolism, growth, and progression. HIF1α is a critical mediator of a response to hypoxia, which binds to HIF1β (also known as ARNT) to form the transcription factor HIF1; this factor induces the expression of diverse genes that are associated with cell metabolism and tumor angiogenesis—a critical process in solid tumor progression and the intratumoral response to hypoxia. VEGFA induces this process, where HIF1 upregulates the expression [23]. HIF1α is not expressed in healthy tissues, but it is expressed in thyroid cancer, especially in more aggressive forms like ATC. Another target of HIF1 is a MET oncogene. It is also important to observe that HIF1 in thyroid cancer is upregulated by the PI3K/Akt and MAPK pathways (Figure 1) (Table 1) [24].

The TSHR signaling pathway plays a fundamental role in the regulation of thyroid cells proliferation, differentiation, and function, as well as the development of the thyroid gland through the TSH receptor (TSHR). TSHR is coupled with the G-protein, and it is linked to a guanine nucleotide that unchains two intracellular signaling pathways, namely, AMP cycloadenililcyclase Gsα-mediated signaling and intracellular Ca^2+^-inositol 1,4,5-triphosphoolipase Cβ-mediated signaling by Gs or G11 (demonstrated in experimental animals). There is a clinical association between elevated TSH levels and an elevated risk of malignant thyroid nodules in men [30]. However, it is likely that the TSH-TSHR system plays a dual role in thyroid tumorigenesis. It is known that the hyperactivation of TSHR signaling leads to a benign hyperfunction (FTA), which suggests that TSHR signaling (the main mechanism that runs Gsα pathway mutation) can act as a protecting factor against the malignant transformation of thyroid cells. Therefore, the TSH-TSHR system not only suppresses malignant transformation, but it also promotes the growth and progression of thyroid cancer once the oncogenic alteration is started (Figure 1) (Table 1) [25,31]. In contrast, some data show that TSH can induce the phosphorylation of p21-activated kinase 4 (PAK4) via TSHR and promote PTC malignization [32].

Moreover, TSHR can be used as a therapeutic target, since TSH estimulation promotes sodium–iodide symporter (NIS) upregulation and radioiodine uptake of the undifferentiated thyroid cells, being important their TSHR expression in the thyroid metastatic cells [33].

### 2.2. Thyroid Cancer Cells (CCs) and Cancer Stem Cells (CSCs)

Stemness is a property that some cell populations possess for the maintenance of tissue homeostasis (adult stem cells), but, when these cell populations undergo deregulation, they can also contribute to cancer initiation or progression. This cell population is referred to as cancer stem cells (CSCs), and these cells’ most important characteristic is their high potential plasticity and resistance to stress factors in the tumoral microenvironment. Moreover, these cells share specific markers of human embryonic stem cells (hESCs) and the adult stem cells [34,35,36]. The question will be to clarify whether thyroid tissue can contain the population of the CSCs or other cells with the characteristics of stem cells instead of external/internal signals or whether normal cells can also participate in their formation through the mechanisms that control cell regulation. In this scenario, several studies have confirmed the existence of the CSCs population and different studies have identified specific miRNA expression in this cell population, with a role in signaling pathways that are involved in tumoral progression and metastasis [37,38,39].

Moreover, side population (SP) cells have also been identified in thyroid. This cell population that correspond to one subtype of hematopoietic stem cells (HSC) was described for the first time in 1996 by Goodell et al., and it is involved in the malignancy of tumors, including the resistance to chemotherapy [40,41]. In addition, they are implicated in the EMT with a high capacity for sphere-forming when they are cultured in soft agar assays [42].

It is known that the epithelial cells can acquire characteristics of CSCs through the epithelial mesenchymal transition (EMT). Several inducers that can change the transcriptional pattern and cellular epigenetic reprogramming control this process. These molecular factors are described below.

Moreover, a clear correlation between cell metabolic reprogramming and EMT pathways has been identified, where lipid metabolism has an important role in this process [43,44].

### 2.3. EMT/MET Processes Related with ETC

EMT is the biological process in which epithelial cells are transformed in cells with a mesenchymal phenotype through multiple biochemical changes [45]. The reverse process is known as the MET. Both are considered to be extremely important in the establishment of the tumors and local or distant metastasis [46,47]. In the EMT/MET, different molecular pathways, different transcription factors, and the expression of specific cell-surface proteins are involved in providing the migratory capacity, invasiveness, and resistance to apoptosis, among other malignancy characteristics, to tumoral cells [45].

The EMT process has been classified, as follows, according to three different contexts: Type 1, which is associated with embryonic development; Type 2, which is essential for tissue regeneration wound healing; and, Type 3, which is linked to cancer progression and metastasis [45,48]. Genetic and epigenetic alterations are potentially responsible for EMT signals originating in the tumor-associated stroma. The loss of epithelial markers and acquisition of mesenchymal markers, such as α-SMA, FSP1, vimentin, and desmin characterizes this process [45,48,49]. In the thyroid, the downregulation of E-cadherin and the upregulation of vimentin expression were found in BRAF-V600E thyroid tumors as compared with normal thyroid tissue (Figure 2) [50].

Several molecules, such as HGF, EGF, PDGF, and TGF-β, have been identified as inducers of the EMT process. One of the most important molecules is TGF-β, and two mechanisms have been proposed for this, namely, Smad-dependent and independent signaling events. In the first case, this pathway can regulate the expression of various transcription factors, which are, as follows: Snail, Slug, zinc finger E-box binding homeobox 1 (ZEB1), Twist, Homeobox protein Goosecoid, and FOXC2. In human and murine thyroid cancer and other carcinomas, the expressions of high levels of Slug, Snail, and Twist and their upregulation are also associated with the transdifferentiation program of thyroid cells into undifferentiated CSC-like cells (Figure 2) [41,47,50]. Moreover, miRNAs play an important role as a regulatory in WDTC and the overexpression of miRNA-146b, miRNA-221, miRNA-222, and miRNA-181b was associated with their malignization and the miRNA-17-92 cluster was identified in ATC [51].

Little is known regarding the MET processes. Some of the authors have suggested that mesenchymal cells and epithelial cells of the tumors engage in cross-talk, facilitating the migration of the latter population (epithelial-like population) and the colonization of other tissues [52,53].

## 3. Tissue and Inflammatory Mediators in Relation to the EMT Process in ETC

Several data support the idea of the strong correlation between chronic inflammation process and carcinogenesis, with the infiltrated inflammatory cells in the tissue having the capacity to release different cytokines that are involved in the cellular malignization [54].

On the other hand, the incidence of thyroid cancer (especially PTC) has rapidly increased over time, in parallel to the increment of obesity prevalence. A meta-analysis of adiposity measures and thyroid cancer risk using 21 articles yielded data on 12,199 thyroid cancer cases; it found an increased risk (25% excess) for the development of thyroid cancer in overweight individuals and a 55% greater risk in obese individuals [55]. In this scenario, epidemiological studies have established an association between obesity, insulin resistance, type 2 diabetes (DM2), and several cancer types, especially breast and colon cancer [56]. Although the association with ETC is not completely understood, several mechanisms have been proposed for this link, which include elevated levels of TSH, insulin, glucose and triglycerides, insulin resistance, obesity, vitamin D deficiency, and antidiabetic medications [57]. However, several reports, including literature reviews and meta-analyses, have provided conflicting results [58]; nevertheless, according to some authors, obesity may contribute to ETC development [59], whereas it has also been associated with a decreased risk of medullar thyroid cancer [56]. Most of the reports do not offer significant evidence regarding the causal mechanisms of these associations. The link among obesity, DM2, dyslipidemia, and the risk of ETC is not completely understood. It appears reasonable to suggest that insulin resistance and hypothetical factors from adipose tissue (AT) may contribute to a higher risk of follicular thyroid cell proliferation [60].

### 3.1. Adipose Tissue

AT is considered to be an inflammatory tissue that is able to express both immuno-receptors and immunoregulatory molecules that promote a pro-tumor immune microenvironment. For these reasons, it has been proposed that obesity as an inflammatory process, being the link between obesity and some solid tumors. The identification in the adipose tissue of macrophagic markers that positively correlated with both adipocyte size and body mass supports this hypothesis, which thereby indicates the association between macrophage infiltration in the tumoral tissue and the development of obesity [61].

The macrophage population is divided into two subgroups (M1 and M2) with different cellular actions, thereby the cytokines interferon α (IFNα) or lipopolysaccharide (LPS) can stimulate M1 and produce proinflammatory cytokines; in contrast, cytokines, such as interleukins (ILs) IL-4, IL-10, or IL-13, can stimulate M2, which are related to humoral immune response and they are able to produce anti-inflammatory lymfokines and also participate in tissue repair [62,63]. These two types of macrophages are responsible for almost all adipose tissue TNFα expression and a significant amount of iNOS and IL-6 expression. Moreover, TNFα has an antiproliferative action in a human PTC cell line through a receptor-mediated mechanism [64]. However, the high TNFα exposure that is provided by obesity may induce TNFα resistance that facilitates tumor progression [65].

On the other hand, AT is a critical regulator of the energy balance and substrate metabolism. It plays an important role in energy homeostasis through the production and secretion of several substances with endocrine or paracrine functions. Furthermore, AT is not only considered as a storage organ, but also as an important endocrine organ that releases various proteins that constitute the tumor microenvironment [66]. Many studies have associated an excessive amount of AT with the development of different diseases, such as DM2, premature atherosclerosis, and cardiovascular disease [67]. In ETC, the biological features of AT could play a crucial role in carcinogenic processes. Moreover, it has been demonstrated that many different molecules that are derived from AT, such as leptin, adiponectin (Acrp30), and cholesterol, promote proliferation and invasion processes via the activation of specific transcription factors.

#### 3.1.1. Leptin and Its Signaling Pathway

Leptin is an important adipokine with a molecular weight of 16 kDa. Its major functions are regulating appetite, energy homeostasis, and lipid metabolism, as well as regulating cell differentiation and proliferation of different types of cells. The LEP gene, which is located on chromosome 7q32.1, encodes it, and the AT mainly secretes it [48,68,69].

Leptin is one of the most important mediators between obesity and increased cancer risk. This protein exerts its effects through binding to the OBR receptor, which is widely expressed in a variety of tissues transducing leptin-mediated downstream signaling events [68]. Leptin and OBR receptors have been reported to be overexpressed in numerous types of cancer. What is more, their expression intensity is associated with proliferation, migration, invasion, EMT, and prognosis in several common types of cancer [47,48].

The LEPR gene encodes the OBR receptor, which is located on chromosome 1p31.3. This receptor belongs to the family of class I cytokine receptors and it presents six isoforms that were generated by alternative splicing (OBRa-OBRf). However, only the long form (OBRrb) contains the intracellular motifs that are required for the activation of the signaling [48].

In ETC, the expression of leptin and OBRs is also increased. In a case-control study by Hedayati et al. [70], the leptin levels were higher in ETC patients when compared with healthy subjects. Moreover, in Cheng et al.’s research [71], leptin and its receptor were overexpressed in PTC cells and associated with an aggressive phenotype, larger tumor size, and nodal metastasis. Finally, leptin was also shown to enhance the migration of PTC cells, whereas it inhibited the migration of anaplastic and follicular CCs [72]. In a study on a cohort of Chinese patients with PTC, the leptin and OBRs were expressed in 72.4% and 73.9% of PTC patients [69]. Similarly, in another study of Uddin et al., OBRs and leptin were expressed in 80.1% and 49.1% of PTC cases, respectively [73]. Thus, the previous studies showed that a proportion of PTC tumors express leptin and OBRs, and their expression is associated with aggressive PTC tumor behaviors, poorer disease-free survival, disease persistence, and recurrence [69,70,73].

#### 3.1.2. Leptin and the EMT Process and ETC

Leptin binds to its receptor through the extracellular region of the OBRrb receptor, which comprises two cytokine homology regions (CHRs). However, only the CHR2 domain is necessary for binding with leptin. Once leptin is bound, the transdimerization of two leptin-OBRrb dimers takes place. This process induces changes in the intracellular region of the OBRrb receptor. Thus, different pathways become activated, which are as follows: the classic cytokine Janus kinase 2/signal transducer and activator of transcription 3 (JAK2/STAT-3) pathway, Ras/extracellular signal-regulated kinases 1/2 (Ras/ERK1/2) signaling cascade, and phosphoinositide 3 kinase/protein kinase B (PI3K/Akt) growth/anti-apoptotic pathway) (Figure 3) [48,68,69].

The intracellular region of the OBRrb contains the three following boxes: a proline-rich region called box 1, which is essential for the binding of the FERM domain of JAK2; a region called box 2 that interacts with the SH2 domain of JAK2; and, box 3, which contains Tyr1077 and Tyr1138, which are necessary residues for the activation of STAT-3 and STAT-5. In addition, JAK2 phosphorylates STAT-3 at Tyr705, inducing its dimerization and translocation to the nucleus, as well as insulin receptor substrate 1 (IRS-1) phosphorylation [74]. Consequently, the PI3K/Akt pathway is activated, which involves the activation of Akt2 and PAK and the phosphorylation of Akt. The latter phosphorylates the GSK-3β, allowing for β-catenin to enter the nucleus and bind to the TCF/LEF complex. Akt2 and PAK phosphorylate Twist and Snail, respectively, promoting their entry into the nucleus. In contrast, JAK2 phosphorylates the Tyr985 of OBRrb, and then the SHP2 protein is anchored and recruits Grb2, which promotes an activation of kinases ERK1/2. Thus, the transcription factors: Slug, Zeb1, and Twist are phosphorylated, entering to the nucleus and promoting the inhibition of the epithelial genes and the overexpression of mesenchymal genes, such as vimentin, cyclin D1, MMP9, VEGF, and COX2, among others (Figure 3) [48,75].

It has been proposed that there are some functional interactions between leptin, metastasis-associated protein 1 (MTA1), and Wnt signaling components [49]. During the EMT process, the expression of Wnt is promoted when the expression of E-cadherin is decreased, and this is associated with high levels of Snail in the nucleus and the nuclear translocation of β-catenin from the cytoplasm, becoming part of the Tcf/LEF complex (Figure 3) [45,68].

In thyroid cancer, there are few studies regarding the relationship between leptin and the EMT process, but some reports have demonstrated that treatment with leptin can promote cell growth and modulated migration of CCs, as well as inhibiting apoptosis through of the upregulation of the XIAP gene [76].

#### 3.1.3. Adiponectin

Acrp30 is the most abundant adipokine that is released by white AT and it circulates oligomers of different molecular weights, which may be of low molecular weight (LMW), medium molecular weight (MMW), or high molecular weight (HMW) [58]. The Acrp30 gene codes for a 244-amino acid polypeptide and it is located on chromosome 3q27 [59]. Its regulation depends on various factors, such as genetic and inflammatory parameters, hormones, or the percentage of body fat and body mass index [47]. Among these, Acrp30 has multiple functions in the regulation of energy homeostasis, insulin sensitivity, and inflammation processes [58].

#### 3.1.4. Adiponectin and the Signaling Pathway

Acrp30 has been considered as a negative regulator of the tumor progression and invasion processes via AdipoR1 and AdipoR2. Furthermore, this molecule plays an important role in increasing apoptosis via the activation of the pAMPK/mTOR pathways (Figure 4) [66,77]. In contrast, other studies have challenged the antitumor role of Acrp30, demonstrating that its presence promotes metastasis and proliferation through the MAPK and NF-κB pathways (Figure 4) [78,79].

The AdipoR1 and AdipoR2 receptors are expressed in macrophages, dendritic cells (DCs), and lymphocytes of various tissues and organs. These two classical Acrp30 receptors share 67% identity of their protein sequence [78]. They belong to the family of seven transmembrane proteins with internal N-terminus and external C-terminus regions, and they may form homo- and hetero-multimers [78,80]. Genes that are situated on the 1p36.13-q41 and 12p13.31 chromosomal regions encode AdipoR1 and AdipoR2, respectively [81].

It seems that Acrp30 exerts antineoplastic effects via two mechanisms: First, it can directly act on CCs by stimulating receptor-mediated signaling pathways. Second, it may act indirectly by modulating insulin sensitivity at the target tissue site, regulating inflammatory responses, and influencing tumor angiogenesis [78]. This review will be focused on the receptor-mediated (AdipoR1 and AdipoR2) signaling pathways.

The evidence suggests that most of the effects of Acrp30 are related to the inhibition of carcinogenesis through AMPK, as well as the inhibition of leptin-induced JAK2 activation and STAT-3 transcriptional activity via increasing the PTP1B protein [78,82]. For example, the activation of cAMP/protein kinase A (PKA), inhibition of β-catenin, and the reduction of reactive oxygen species (ROS) have also been implicated in CCs response to Acrp30 (Figure 4) [78]. Conversely, new molecular data suggest that Acrp30 and its receptor promote proangiogenic effects that could promote tumor growth. This event could be promoted by ceramidase activity from the classical Acrp30 adiponectin receptors [78,83].

Acrp30 binds to AdipoR1 or AdipoR2, initiating a downstream signaling pathway [82]. Bax is induced, thereby promoting apoptosis. Moreover, Ser/Thr kinase LKB1, calcium-dependent kinases (CaMKK), and the adaptor protein APPL-1 are cofactors that activate AMPK, and APPL-1 acts as a signaling pathway mediator in cross-talk with Acrp30 and insulin, which directly interacts with insulin receptor substrates [78,81]. Furthermore, activated AMPK stimulates JNK, PP2A, and p53/p21/p27 (cell cycle regulators) and inhibits fatty acid synthase (FAS)/ACC and the mTOR/S6K axis through TSC2 phosphorylation [81,84]. These factors result in reduced fatty acid synthesis; decreased cellular growth, proliferation, and DNA mutagenesis; and, increased cell cycle arrest and apoptosis, thereby negatively influencing carcinogenesis (Figure 4) [81].

Acrp30 has a negative effect on PI3K/Akt activation. Thus, Akt, which promotes cell proliferation and survival, remains inactivated. PTEN acts as an inhibitor of PI3K signaling, and cell line-specific PTEN deficiency may determine whether AMPK or Akt prevails in regulating mTOR activation. Furthermore, Acrp30 inhibits thioredoxin and the thioredoxin reductase-mediated inhibition of PTEN, thereby influencing PI3K/Akt signaling (Figure 4) [78].

The mediators of the MAPK, cascade, such as JNK, p38, and ERK1/2, can also be related to Acrp30. In different types of cancers, treatment with Acrp30 led to increased JNK activity, resulting in Caspase-3-mediated apoptosis. It seems that AMPK plays a crucial role in JNK phosphorylation. Furthermore, Acrp30 treatment resulted in decreased levels of c-myc, cyclin D, and Bcl-2, while increasing the expression of p53 and Bax, thereby promoting cell cycle arrest and apoptosis (Figure 4) [78]. In contrast, it has been observed that Acrp30 can increase p38 activity, positively influencing the carcinogenesis processes [83].

#### 3.1.5. Adiponectin, the EMT Process and ETC

An EPIC cohort’s study that was published by Dossus et al. [77] compared the levels of various proteins that are associated with inflammation (IL-10) and Acrp30 with ETC risk. The researchers established that women with high Acrp30 levels were less likely to develop ETC, while those with high IL-10 levels had increased TC risk. A few studies suggest that Acrp30 may have a role in EMT, affecting the expression of molecules, like cytokines, vimentin, or E-cadherin. Nigro et al. [80] demonstrated that Acrp30 promotes the apoptosis and oxidative stress in colorectal CCs and increases the pro-inflammatory IL-6 and IL-8 cytokines, which thereby decreases cell survival and migration processes, while reversing the EMT processes [80,85].

In relation to MMPs and tissue inhibitor of MMPs (TIMPs), their role in inducing EMT has been well reported, as well as their part in the migration and invasion processes in ETC [86,87,88]. It has been suggested that ETC may secrete a factor(s) that stimulates MMPs or TIMPs surrounding tissues. Patel et al. demonstrated that the expression of MMPs and TIMPs in recurrent PTC was lower in TIMP when compared with non-recurrent PTC (*p* = 0.049). Thus, MMPs and TIMPs are all expressed in ETC and could be important in promoting recurrence [86,87,88]. Moreover, it has been reported that Acrp30 can decrease MMP activity via the secretion of human monocyte-derived macrophages via the Syk pathway [85]. Thus, it is well established that tumor cells can induce tumor-associated macrophages (TAMs) that express immunosuppressive cytokine association [66]. Sun et al. [89] revealed that Acrp30 deficiencies promote tumor progression, which reduces the infiltration of macrophages in tumor tissues.

In ETC, Cheng et al. [90] found that the protein levels of the heptahelical transmembrane adiponectin receptors 1 (AdipoR1) and 2 (AdipoR2) modulated by the histone acetylation pattern were increased in some TC patients. This overexpression, which was located in some PTC tissues, was associated with a better prognosis. Moreover, Mitsiadis et al. [91] also reported an inverse relationship of circulating Acrp30 levels with ETC risk.

### 3.2. Plasma Lipoprotein Particles

Significant evidence supports the key role of lipid pathways in cancer development [92,93]. Tumor cells require an increased supply of lipids, which can be obtained by either increasing the exogenous lipids uptake or by upregulating their endogenous synthesis [94,95,96,97]. CCs can take up exogenous cholesterol from low-density lipoproteins (LDL) or be exported into the circulation as high-density lipoproteins (HDLs) to regulate their cholesterol requirement. When cholesterol is in excess, it can be exported into the circulation or locally stored in lipid droplets. Stored cholesterol and one of the main oxidized derivatives of cholesterol, 27-hydroxycholesterol (27-HC), are considered as one of the hallmarks of cancer aggressiveness [98,99]. Alterations in the plasma membrane composition influence the signaling pathways that are associated with EMT processes [44]. Indeed, the lipid composition of epithelial cells in EMT processes are distinct, and they exhibit a significant increase in sphingomyelin and diacylglycerol contents, whereas cholesterol is markedly reduced, which thereby contributes to the enhancement in membrane fluidity and the conversion to the mesenchymal phenotype [100]. However, the statin-induced reduction in intracellular cholesterol levels impaired the cell motility and the metastatic potential in CCs undergoing EMT [101]. These divergent findings suggest a complex relationship between cholesterol, tumor proliferation, and EMT. Cholesterol affects tumor proliferation and metastasis through a range of mechanisms, and it is likely that it plays both promoting and inhibiting roles throughout tumor development. Beyond cholesterol, fatty acid synthesis may also promote the pro-tumorigenic pathways that are involved in EMT-induced cell motility [102]. Indeed, breast CCs undergoing EMT showed an increased expression of FAS and accumulations of saturated fatty acids, thereby regulating lipid rafts’ organization and activating EMT-inducer VEGF/VEGFR2 signaling [102]. FAS inhibition prevented EMT execution, migration, and invasion induced by the hepatocyte growth factor in breast CCs, which is in line with these findings [103]. However, these findings have not been reproduced in other CCs undergoing EMT [104].

#### 3.2.1. Role of Low-Density Lipoproteins

The LDL receptor (LDLR) gene resides on chromosome 19 at the band 19p13.2, which encodes for a cell-surface receptor of 839 amino acids [105]. Increased LDLR expression and, consequently, the uptake of LDL cholesterol has been reported in breast CCs [106,107,108]. In line with these findings, LDL promoted breast CC proliferation, migration, and loss of adhesion, which are hallmarks of the EMT. Furthermore, adhesion molecules, such as cadherin-related family member 3, CD226, Claudin 7, and Ocludin, were downregulated in breast cancer cells exposure to LDL [109]. In this context, the accumulation of 27-HC induced the activation of STAT-3, which promoted the angiogenesis of breast CCs via the ROS/STAT-3/VEGF47 and induced EMT [110]. This mechanism may also contribute to promoting migration and invasion, via STAT-3/MMP-9 and STAT-3/EMT signaling (Figure 5) [111].

Under conditions of oxidative stress, LDL can be oxidatively modified, which results in the formation of lipid peroxidation metabolites, which are closely associated with carcinogenic processes [112]. The levels of serum oxidized LDL (oxLDL) were associated with an increased risk of breast cancer [113], which, in turn, triggers pro-oncogenic signaling in breast CCs [114]. OxLDL lecithin-like receptor 1 (LOX-1) is the main receptor for the internalization of oxLDL. LOX-1 is located in 12p13.2, which encodes a 273-amino acid protein with a short N-terminal intracellular domain, a transmembrane domain, and an extracellular region, followed by a C-type lectin-like domain [115]. Importantly, oxLDL enhanced the LOX-1 expression and lipid accumulation in tubular epithelial cells, which induced the formation of ROS and EMT through ROS [116]. When LOX-1 was overexpressed in prostate CCs and induced by oxLDL, it promoted EMT by downregulating E-cadherin and plakoglobin, whereas it upregulated vimentin, N-cadherin, MMP-2, and MMP-9 (Figure 5) [116].

These studies show the importance of LDL-related pathways in controlling EMT processes and cancer aggressiveness. However, the impact of LDL and oxLDL on EMT processes that are related with thyroid CCs is unknown.

#### 3.2.2. Role of High-Density Lipoproteins

A few studies have addressed the relationship between HDL transporters and EMT in CCs, with divergent results. The gene SCARB1, located in 12q24.31, encodes the scavenger receptor class B, type I (SR-BI), which is the main HDL receptor. This integral membrane protein has 509 amino acids and contains N- and C-terminal cytoplasmic domains, two transmembrane domains, and a large extracellular domain that contains multiple sites for N-linked glycosylation [117]. SR-BI was found to be highly expressed in human metastatic melanoma and regulated EMT processes in melanoma cells [118]. SR-BI downregulation resulted in the loss of the EMT signature in close association with protein glycosylation, including STAT-5, prevented the migration and invasion of melanoma cells, and reduced xenograft tumor growth. However, the SR-BI-mediated effects on EMT were not related to its ability to transport HDL cholesterol [118].

The ABCA1 gene, located in 9q31.1, encodes the ATP binding cassette (ABC) A1 (ABCA1), which is the main cholesterol efflux transporter from cells to HDL. It is ubiquitously expressed as a 220-kDa protein, and it can activate multiple signaling pathways, including the JAK/STAT, PKA, and PKC pathways [119]. Indeed, ABCA1 was significantly upregulated in an EMT of different metastatic CCs [120]. Furthermore, CCs that were induced to undergo an EMT and treated with a range of antimetastatic drugs showed significant ABCA1 downregulation [120]. These results strongly indicate that the ABCA1-mediated effects on plasma membrane cholesterol content and fluidity are critical determinants of their metastatic capacity. In addition, ABCA1 overexpression leads to EMT and the increased invasion of colorectal CCs [121]. However, ABCA1 also inhibits the development of tumors by inhibiting cellular proliferation [122]. ABCA1 reduces mitochondrial cholesterol and ultimately supports CC survival, thereby highlighting the dual role of cholesterol in regulating proliferation and metastasis (Figure 5) [122].

It is noteworthy that apoAI overexpression or exogenous human apoAI, which is the main HDL protein, reduced the malignant properties of colorectal CCs are driven by ABCA1 overexpression [121]. HDL mainly transports sphingosine-1-phosphate (S1P); S1P activates the PI3K/Akt and ERK1/2 signaling pathways, resulting in the MMP-7 upregulation of hepatocellular carcinoma [123]. MMP-7 mediates the shedding of syndecan-1, which, in turn, causes an increase in TGF-β1 production and induces EMT [123]. Based on these results, treatments with some specific HDL components could be considered as potential therapies for treating some cancer invasiveness. However, the effects of lipoproteins and their specific components on EMT processes related to thyroid CCs remain completely unknown.

## 4. Conclusions

The incidence of ETC, especially PTC, has rapidly increased in recent decades, in parallel with the increase in the prevalence of obesity and other inflammatory diseases. Some reports indicate that inflammatory mediators, such as adipokines, can contribute to cell proliferation and the dedifferentiation of the follicular thyroid epithelium, especially in circumstances of greater genetic susceptibility. These cytokines can induce EMT processes in human PTC cell lines; furthermore, the link between inflammation and tumor metastasis progression in these types of tumors has been demonstrated. However, significant evidence has not been offered regarding the causal mechanisms of these associations and the link between these pathologies. As of yet, neither the risk of ETC nor the molecular mechanism that regulates both EMT and MET are completely understood.

In conclusion, inflammatory mediators, like adipokines and metabolite of cholesterol, can contribute to promoting the aggressive behaviours of thyroid tumors, and the molecular mechanism that controls this process should be investigated.

## Figures and Tables

**Figure 1 ijms-20-02466-f001:**
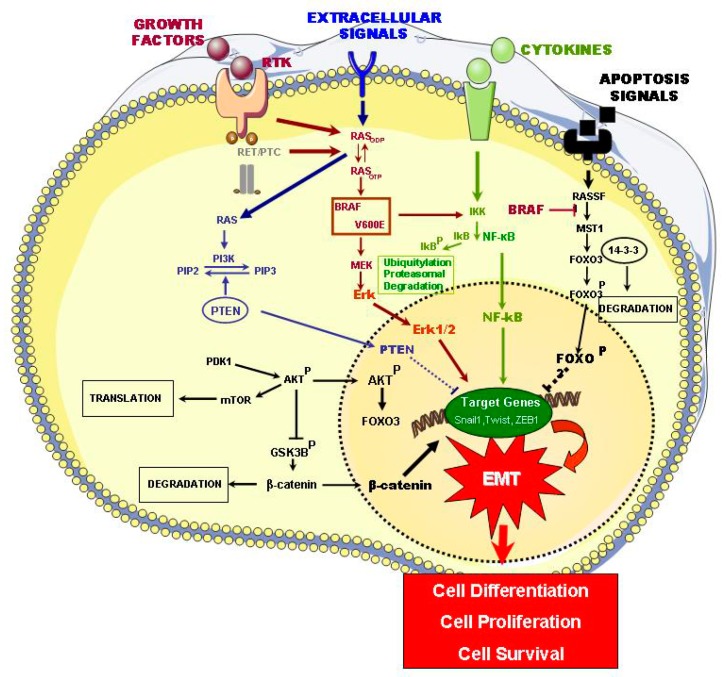
Schematic representation of relevant signaling pathways that are involved in epithelial thyroid carcinogenesis (ETC) and their link with epithelial mesenchymal transition (EMT) process. Part of Servier Medical Art by Servier is licensed under a Creative Commons Attribution 3.0 Unported License. (https://smart.servier.com/image-set-download).

**Figure 2 ijms-20-02466-f002:**
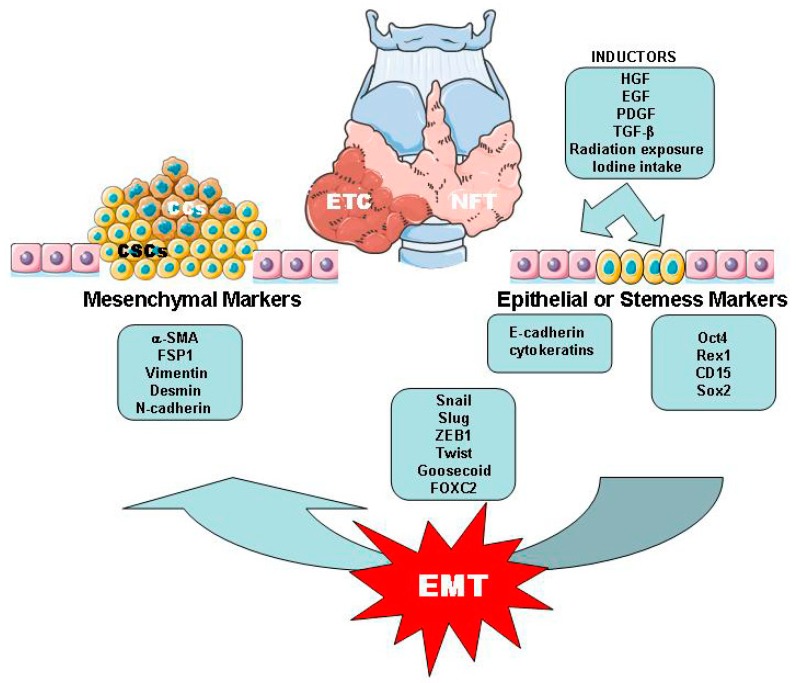
Schematic representation of the most significant signaling inductors involved on the cell dedifferentiation of adult stem cells and normal follicular cells by EMT process. EMT, thyroid cancer cells (CCs) and cancer stem cells (CSCs). Part of Servier Medical Art by Servier is licensed under a Creative Commons Attribution 3.0 Unported License. (https://smart.servier.com/image-set-download).

**Figure 3 ijms-20-02466-f003:**
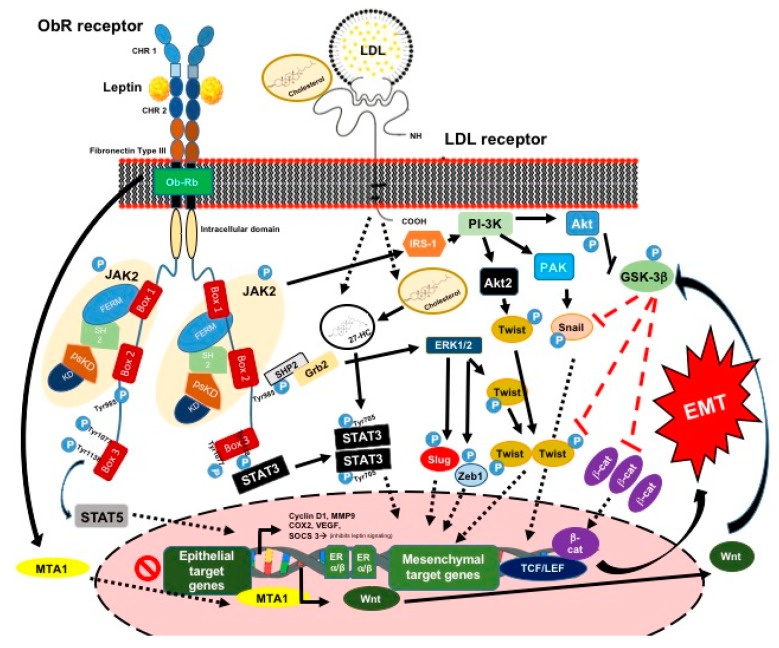
Schematic representation of leptin-induced signaling pathway. Leptin, cholesterol and low-density lipoprotein (LDL) particles interact with their respective receptors, which trigger a cascade of signaling events promoting EMT process.

**Figure 4 ijms-20-02466-f004:**
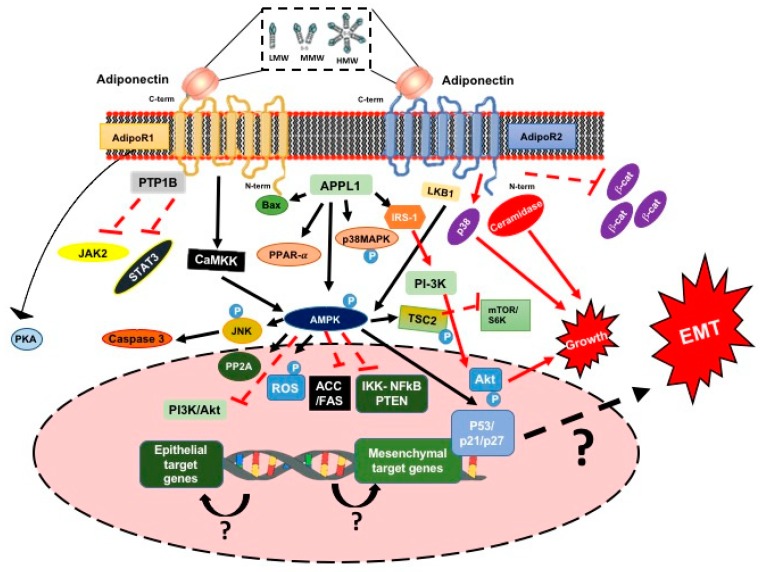
Schematic representation of adiponectin-induced signaling pathway. Adiponectin interacts with its respective receptors, AdipoR1 and AdipoR2, in order to activate a cascade of signaling events. Further studies are needed in order to clarify the possible activation of EMT processes induced by adiponectin.

**Figure 5 ijms-20-02466-f005:**
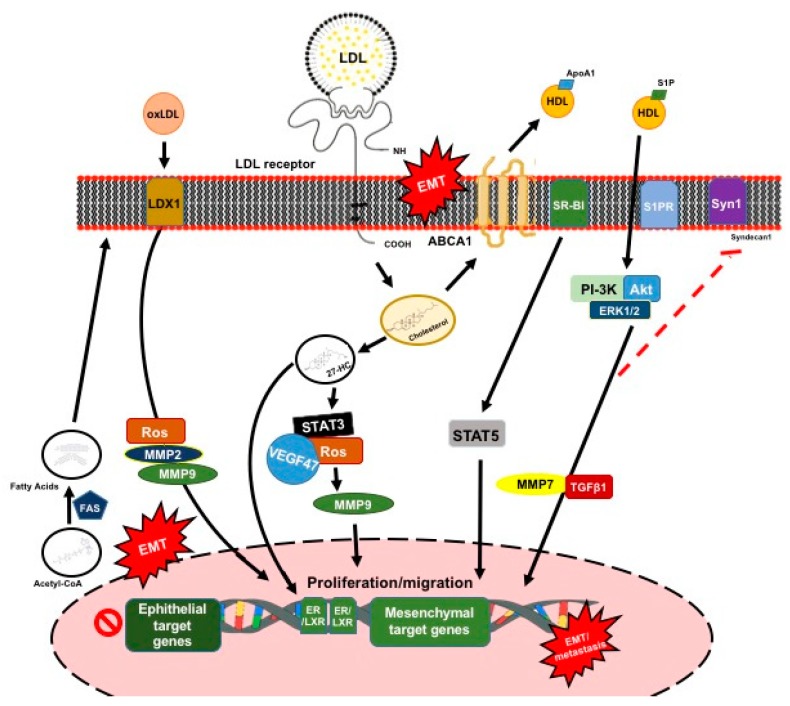
Schematic representation of mechanisms by which LDL, the oxidized form of LDL, high-density lipoprotein (HDL) and their specific receptors induce proliferation, migration, and EMT processes in cancer cells. These lipoprotein particles interact with their respective receptors, which trigger a cascade of signaling events promoting EMT process. The cholesterol that is transported by the lipoproteins and the synthesis of fatty acids play a critical role in these signaling pathways.

**Table 1 ijms-20-02466-t001:** Main signaling pathways and gene mutations involved in thyroid carcinogenesis.

Signaling Pathways Affected	Mutations	Types of Thyroid Tumors	Refs
**MAPK**	BRAF V600E, RAS, RET/PTC, RTK, ALK	PTC	[11,14]
	PI3K/Akt1, PTEN	FTC	[11]
	TGF-β1	PTC, FTC, ATC, PDTC	[16]
**PI3/AKT**	PTEN	FTA, FTC	[17]
	Akt1, Akt2	FTC	[17]
**NF-KB**	RET-PTC, RAS, BRAF-V600E	PTC, FTC, ATC	[18,19,20]
**RASSF1/MST1/FOXO3**	MEK/MAPK, RASSF1/MST1/FOXO3, NF-κB, BRAF-V600E, ERK 1/2, Akt	PTC, FTC, ATC, PDTC	[21]
**WNT/β-CATENIN**	CTNNB1	ATC, PDTC	[22]
**HIF1A**	HIF1, VEGFA, MET	PDTC, FCT, ATC	[23,24]
	HIF1α	ATC	[19]
**TSHR**	TSH-TSHR, NIS	PDTC, FCT, ATC, PTC	[25]
	PAK4	PTC	[25]

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
