# Peer review of "Cross-Talk between Inflammatory Mediators and the Epithelial Mesenchymal Transition Process in the Development of Thyroid Carcinoma"

_ijms, 2019, doi:10.3390/ijms20102466_

Reviewer 1 Report

This is an interesting review, well written and giving interesting insights about the existing cross talk between inflammatory mediators and EMT in the context of thyroid carcinoma. 

The part about Cancer Stem Cells need to be more developed as the molecular factors that are involved are not clearly described.

Author Response

Reviewer #1

This is an interesting review, well written and giving interesting insights about the existing cross talk between inflammatory mediators and EMT in the context of thyroid carcinoma.

The part about Cancer Stem Cells needs to be more developed as the molecular factors that are involved are not clearly described.

We are grateful the Reviewer for his/her comments and suggestions which have helped us to significantly improve the manuscript. The cancer stem cells part has been more developed. The molecular factors involved in these processes have been included in section 2.3. The changes have been highlighted in the revised version of the manuscript.

Reviewer 2 Report

The review by Revilla et al. is focused on the effects that the   complex network between inflammatory processes, metabolic mediators and EMT has on the development and progression of the thyroid carcinoma, even if many facts discussed by the authors affect phenomena shared by many cancer types.

The topic of this review is very relevant as testified by the many papers covering these issues published in the last few years.

Some sections of the review are very detailed, but in some cases, in my opinion, some relevant results are not discussed at all. In addition, the list of references should be updated: 80% of the papers cited has been published before 2016.

Specific comments

 Title …Epithelial Mesenchymal Transition/Mesenchymal Epithelial Transition processes

As the authors state at line 261 “Little is known about the MET processes. Consistently, nothing is discussed about topic. Hence, there is no reason to mention this obscure phenomenon in the title.

 Lines 64-65: stages I and II, stage IV disease

Indicate briefly the features of the different stages of thyroid carcinoma (for the benefit of non-pathologist readers).

Lines 90-92such tumors arise…in benign lesion or in differentiated carcinoma

The transformation of papillary carcinoma tumors into undifferentiated and

aggressive anaplastic carcinoma is an extremely relevant problem for medical research. In recent years some mutations/alterations specifically associated to anaplastic transformation have been identified. These data should be reported to the readers.

Line 178: “especially in terms of CAT and CPDT

These two abbreviations are neither present in the list of abbreviations nor explained in the text.

Section 2.1: Signaling pathways of ETC oncogenesis

In this section the Authors discuss well-established data on the signaling cascades involved in oncogenesis. According to the relevance of omics studies in oncology, it could be very useful to report the common genetic alterations associated with thyroid carcinoma tissues. In addition, the reader could be interested in having some information on both the gene expression profiling of the pathological thyroid and the dysregulation of tissue and /or circulating miRNA in patients.

Line 197: TSHR signaling can act as a protecting factor against the malignant...”

This specific statement should be well discussed, and appropriate references should be reported since TSH/TSHR signaling is usually considered involved in tumor growth.

Lines 201-2017:In this scenario…thyroid cell proliferation”

These two paragraphs could be more appropriately moved in the section 3.1. Adipose tissue

Line 228: side population (SP)

It could useful to define the concept of “side population cell” and appropriate references should be reported.

Section 3: Tissue and inflammatory mediators…

The role of inflammatory cytokines in tumors is relevant. A specific paragraph on this topic could be useful.   

Ref 59

The list of authors is wrong.

Author Response

Reviewer #2:

Some sections of the review are very detailed, but in some cases, in my opinion, some relevant results are not discussed at all. In addition, the list of references should be updated: 80% of the papers cited has been published before 2016.

We are grateful for your critique and suggestions, which have helped us to significantly improve the manuscript. Our responses are presented below, and the changes have been highlighted in the revised version of the manuscript. We have addressed most of your concerns and include new and recent relevant references, but in some specific topics, no recent papers have been published.

 Specific comments

 Title …Epithelial Mesenchymal Transition/Mesenchymal Epithelial Transition processes… As the authors state at line 261 “Little is known about the MET processes. Consistently, nothing is discussed about topic. Hence, there is no reason to mention this obscure phenomenon in the title.

We have changed the title

Lines 64-65: stages I and II, stage IV disease Indicate briefly the features of the different stages of thyroid carcinoma (for the benefit of non-pathologist readers).

We have clarified this point

Lines 90-92 “such tumors arise…in benign lesion or in differentiated carcinoma” The transformation of papillary carcinoma tumors into undifferentiated and aggressive anaplastic carcinoma is an extremely relevant problem for medical research. In recent years some mutations/alterations specifically associated to anaplastic transformation have been identified. These data should be reported to the readers.

We have summarized mutations/alterations specifically associated to anaplastic transformation in Table 1

Line 178: “especially in terms of CAT and CPDT” These two abbreviations are neither present in the list of abbreviations nor explained in the text.

These abbreviations have been included

Section 2.1: Signaling pathways of ETC oncogenesis In this section the Authors discuss well-established data on the signaling cascades involved in oncogenesis. According to the relevance of omics studies in oncology, it could be very useful to report the common genetic alterations associated with thyroid carcinoma tissues. In addition, the reader could be interested in having some information on both the gene expression profiling of the pathological thyroid and the dysregulation of tissue and /or circulating miRNA in patients.

As commented above, we have summarized mutations/alterations specifically associated to thyroid carcinoma tissues in Table 1. We have also included a brief description of the regulatory role of miRNAs in thyroid cancer in section 2.3

Line 197: TSHR signaling can act as a protecting factor against the malignant...” This specific statement should be well discussed, and appropriate references should be reported since TSH/TSHR signaling is usually considered involved in tumor growth.

We have included this point in the last paragraph of section 2.1

Lines 201-2017: “In this scenario…thyroid cell proliferation”

These two paragraphs could be more appropriately moved in the section 3.1. Adipose tissue

These paragraphs have been moved to section 3.1

Line 228: side population (SP) It could useful to define the concept of “side population cell” and appropriate references should be reported.

SP has been now defined

Section 3: Tissue and inflammatory mediators… The role of inflammatory cytokines in tumors is relevant. A specific paragraph on this topic could be useful. 

A specific paragraph has been included in section 3.1

Ref 59 The list of authors is wrong.

This reference has been corrected

Round  2

Reviewer 2 Report

The authors have addressed all the questions.